# Sobolev Independence Criterion

**Youssef Mroueh, Tom Sercu, Mattia Rigotti, Inkit Padhi, Cicero Dos Santos** *
IBM Research & MIT-IBM Watson AI lab
mroueh,mrigotti@us.ibm.com,inkit.padhi@ibm.com

## Abstract

We propose the Sobolev Independence Criterion (SIC), an interpretable dependency measure between a high dimensional random variable $X$ and a response variable $Y$. SIC decomposes to the sum of feature importance scores and hence can be used for nonlinear feature selection. SIC can be seen as a gradient regularized Integral Probability Metric (IPM) between the joint distribution of the two random variables and the product of their marginals. We use sparsity inducing gradient penalties to promote input sparsity of the critic of the IPM. In the kernel version we show that SIC can be cast as a convex optimization problem by introducing auxiliary variables that play an important role in feature selection as they are normalized feature importance scores. We then present a neural version of SIC where the critic is parameterized as a homogeneous neural network, improving its representation power as well as its interpretability. We conduct experiments validating SIC for feature selection in synthetic and real-world experiments. We show that SIC enables reliable and interpretable discoveries, when used in conjunction with the holdout randomization test and knockoffs to control the False Discovery Rate. Code is available at http://github.com/ibm/sic.

## 1   Introduction

Feature Selection is an important problem in statistics and machine learning for interpretable predictive modeling and scientific discoveries. Our goal in this paper is to design a dependency measure that is interpretable and can be reliably used to control the False Discovery Rate in feature selection. The mutual information between two random variables $X$ and $Y$ is the most commonly used dependency measure. The mutual information $I(X;Y)$ is defined as the Kullback-Leibler divergence between the joint distribution $p_{xy}$ of $X, Y$ and the product of their marginals $p_x p_y$, $I(X;Y) = \text{KL}(p_{xy}, p_x p_y)$. Mutual information is however challenging to estimate from samples, which motivated the introduction of dependency measures based on other $f$-divergences or Integral Probability Metrics [1] than the KL divergence. For instance, the Hilbert-Schmidt Independence Criterion (HSIC) [2] uses the Maximum Mean Discrepancy (MMD) [3] to assess the dependency between two variables, i.e. $\text{HSIC}(X, Y) = \text{MMD}(p_{xy}, p_x p_y)$, which can be easily estimated from samples via Kernel mean embeddings in a Reproducing Kernel Hilbert Space (RKHS) [4]. In this paper we introduce the Sobolev Independence Criterion (SIC), a form of gradient regularized Integral Probability Metric (IPM) [5, 6, 7] between the joint distribution and the product of marginals. SIC relies on the statistics of the gradient of a witness function, or critic, for both (1) defining the IPM constraint and (2) finding the features that discriminate between the joint and the marginals. Intuitively, the magnitude of the average gradient with respect to a feature gives an importance score for each feature. Hence, promoting its sparsity is a natural constraint for feature selection.

The paper is organized as follows: we show in Section 2 how sparsity-inducing gradient penalties can be used to define an interpretable dependency measure that we name Sobolev Independence Criterion

(SIC). We devise an equivalent computational-friendly formulation of SIC in Section 3, that gives rise to additional auxiliary variables $\eta_j$. These naturally define normalized feature importance scores that can be used for feature selection. In Section 4 we study the case where the SIC witness function $f$ is restricted to an RKHS and show that it leads to an optimization problem that is jointly convex in $f$ and the importance scores $\eta$. We show that in this case SIC decomposes into the sum of feature scores, which is ideal for feature selection. In Section 5 we introduce a Neural version of SIC, which we show preserves the advantages in terms of interpretability when the witness function is parameterized as a homogeneous neural network, and which we show can be optimized using stochastic Block Coordinate Descent. In Section 6 we show how SIC and conditional Generative models can be used to control the False Discovery Rate using the recently introduced Holdout Randomization Test [8] and Knockoffs [9]. We validate SIC and its FDR control on synthetic and real datasets in Section 8.

## 2 Sobolev Independence Criterion: Interpretable Dependency Measure

**Motivation: Feature Selection.** We start by motivating gradient-sparsity regularization in SIC as a mean of selecting the features that maintain maximum dependency between two randoms variable $X$ (the input) and $Y$ (the response) defined on two spaces $\mathcal{X} \subset \mathbb{R}^{d_x}$ and $\mathcal{Y} \subset \mathbb{R}^{d_y}$ (in the simplest case $d_y = 1$). Let $p_{xy}$ be the joint distribution of $(X, Y)$ and $p_x, p_y$ be the marginals of $X$ and $Y$ resp. Let $D$ be an Integral Probability Metric associated with a function space $\mathscr{F}$, i.e for two distributions $p, q$:

$$D(p, q) = \sup_{f \in \mathscr{F}} \mathbb{E}_{x \sim p} f(x) - \mathbb{E}_{x \sim q} f(x).$$

With $p = p_{xy}$ and $q = p_x p_y$ this becomes a generalized definition of Mutual Information. Instead of the usual KL divergence, the metric $D$ with its witness function, or critic, $f(x, y)$ measures the distance between the joint $p_{xy}$ and the product of marginals $p_x p_y$. With this generalized definition of mutual information, the feature selection problem can be formalized as finding a sparse selector or gate $w \in \mathbb{R}^{d_x}$ such that $D(p_{w \odot x, y}, p_{w \odot x} p_y)$ is maximal [10, 11, 12, 13] , i.e. $\sup_{w, \|w\|_{\ell_0} \le s} D(p_{w \odot x, y}, p_{w \odot x} p_y)$, where $\odot$ is a pointwise multiplication and $\|w\|_{\ell_0} = \#\{j | w_j \ne 0\}$. This problem can be written in the following penalized form:

$$\text{(P)}: \quad \sup_w \sup_{f \in \mathscr{F}} \mathbb{E}_{p_{xy}} f(w \odot x, y) - \mathbb{E}_{p_x p_y} f(w \odot x, y) - \lambda \|w\|_{\ell_0}.$$

We can relabel $\tilde{f}(x, y) = f(w \odot x, y)$ and write (P) as: $\sup_{\tilde{f} \in \tilde{\mathscr{F}}} \mathbb{E}_{p_{xy}} \tilde{f}(x, y) - \mathbb{E}_{p_x p_y} \tilde{f}(x, y)$, where $\tilde{\mathscr{F}} = \{\tilde{f} | \tilde{f}(x, y) = f(w \odot x, y) | f \in \mathscr{F}, \|w\|_{\ell_0} \le s\}$. Observe that we have: $\frac{\partial \tilde{f}}{\partial x_j} = w_j \frac{\partial f(w \odot x, y)}{\partial x_j}$. Since $w_j$ is sparse the gradient of $\tilde{f}$ is sparse on the support of $p_{xy}$ and $p_x p_y$. Hence, we can reformulate the problem (P) as follows:

$$\text{(SIC)}: \quad \sup_{f \in \mathscr{F}} \mathbb{E}_{p_{xy}} f(x, y) - \mathbb{E}_{p_x p_y} f(x, y) - \lambda P_S(f),$$

where $P_S(f)$ is a penalty that controls the sparsity of the gradient of the witness function $f$ on the support of the measures. Controlling the nonlinear sparsity of the witness function in (SIC) via its gradients is more general and powerful than the linear sparsity control suggested in the initial form (P), since it takes into account the nonlinear interactions with other variables. In the following Section we formalize this intuition by theoretically examining sparsity-inducing gradient penalties [14].

**Sparsity Inducing Gradient Penalties.** Gradient penalties have a long history in machine learning and signal processing. In image processing the total variation norm is used for instance as a regularizer to induce smoothness. Splines in Sobolev spaces [15], and manifold learning exploit gradient regularization to promote smoothness and regularity of the estimator. In the context of neural networks, gradient penalties were made possible through double back-propagation introduced in [16] and were shown to promote robustness and better generalization. Such smoothness penalties became popular in deep learning partly following the introduction of WGAN-GP [17], and were used as regularizer for distance measures between distributions in connection to optimal transport theory [5, 6, 7]. Let $\mu$ be a dominant measure of $p_{xy}$ and $p_x p_y$ the most commonly used gradient penalties is

$$\Omega_{L^2}(f) = \mathbb{E}_{(x,y) \sim \mu} \|\nabla_x f(x, y)\|^2.$$

While this penalty promotes smoothness, it does not control the desired sparsity as discussed in the previous section. We therefore elect to instead use the nonlinear sparsity penalty introduced in [14] :

$\Omega_{\ell_0}(f) = \#\{j | \mathbb{E}_{(x,y)\sim\mu} \left| \frac{\partial f(x,y)}{\partial x_j} \right|^2 \neq 0\}$, and its relaxation :

$$\Omega_S(f) = \sum_{j=1}^{d_x} \sqrt{\mathbb{E}_{(x,y)\sim\mu} \left| \frac{\partial f(x,y)}{\partial x_j} \right|^2}.$$

As discussed in [14], $\mathbb{E}_{(x,y)\sim\mu} \left| \frac{\partial f(x,y)}{\partial x_j} \right|^2 = 0$ implies that $f$ is constant with respect to variable $x_j$, if the function $f$ is continuously differentiable and the support of $\mu$ is connected. These considerations motivate the following definition of the *Sobolev Independence Criterion* (SIC):

$$\text{SIC}_{(L_1)^2}(p_{xy}, p_x p_y) = \sup_{f \in \mathscr{F}} \mathbb{E}_{p_{xy}} f(x,y) - \mathbb{E}_{p_x p_y} f(x,y) - \frac{\lambda}{2} (\Omega_S(f))^2 - \frac{\rho}{2} \mathbb{E}_\mu f^2(x,y).$$

Note that we add a $\ell_1$-like penalty ($\Omega_S(f)$) to ensure sparsity and an $\ell_2$-like penalty ($\mathbb{E}_\mu f^2(x,y)$) to ensure stability. This is similar to practices with linear models such as Elastic net.

Here we will consider $\mu = p_x p_y$ (although we could also use $\mu = \frac{1}{2}(p_{xy} + p_x p_y)$). Then, given samples $\{(x_i, y_i), i = 1, \dots, N\}$ from the joint probability distribution $p_{xy}$ and *iid* samples $\{(x_i, \tilde{y}_i), i = 1, \dots, N\}$ from $p_x p_y$, SIC can be estimated as follows:

$$\widehat{\text{SIC}}_{(L_1)^2}(p_{xy}, p_x p_y) = \sup_{f \in \mathscr{F}} \frac{1}{N} \sum_{i=1}^N f(x_i, y_i) - \frac{1}{N} \sum_{i=1}^N f(x_i, \tilde{y}_i) - \frac{\lambda}{2} \left( \hat{\Omega}_S(f) \right)^2 - \frac{\rho}{2} \frac{1}{N} \sum_{i=1}^N f^2(x_i, \tilde{y}_i),$$

where $\hat{\Omega}_S(f) = \sum_{j=1}^{d_x} \sqrt{\frac{1}{N} \sum_{i=1}^N \left| \frac{\partial f(x_i, \tilde{y}_i)}{\partial x_j} \right|^2}$.

**Remark 1.** *Throughout this paper we consider feature selection only on $x$ since $y$ is thought of as the response. Nevertheless, in many other problems one can perform feature selection on $x$ and $y$ jointly, which can be simply achieved by also controlling the sparsity of $\nabla_y f(x,y)$ in a similar way.*

## 3    Equivalent Forms of SIC with $\eta$-trick

As it was just presented, the SIC objective is a difficult function to optimize in practice. First of all, the expectation appears after the square root in the gradient penalties, resulting in a non-smooth term (since the derivative of square root is not continuous at 0). Moreover, the fact that the expectation is inside the nonlinearity introduces a gradient estimation bias when the optimization of the SIC objective is performed using stochastic gradient descent (i.e. using mini-batches). We alleviate these problems (non-smoothness and biased expectation estimation) by making the expectation linear in the objective thanks to the introduction of auxiliary variables $\eta_j$ that will end up playing an important role in this work. This is achieved thanks to a variational form of the square root that is derived from the following Lemma (which was used for a similar purpose as ours when alleviating the non-smoothness of mixed norms encountered in multiple kernel learning and group sparsity norms):

**Lemma 1** ([18],[19]). *Let $a_j, j = 1 \dots d$, $a_j > 0$ we have:* $\left( \sum_{j=1}^d \sqrt{a_j} \right)^2 = \inf\{\sum_{j=1}^d \frac{a_j}{\eta_j} :$ $\eta, \eta_j > 0 \sum_{j=1}^d \eta_j = 1\}$, *optimum achieved at $\eta_j = \sqrt{a_j} / \sum_j \sqrt{a_j}$.*

We alleviate first the issue of non smoothness of the square root by adding an $\varepsilon \in (0,1)$, and we define: $\Omega_{S,\varepsilon} = \sum_{j=1}^{d_x} \sqrt{\mathbb{E}_{(x,y)\sim\mu} \left| \frac{\partial f(x,y)}{\partial x_j} \right|^2 + \varepsilon}$. Using Lemma 1 the nonlinear sparsity inducing gradient penalty can be written as :

$$(\Omega_{S,\varepsilon}(f))^2 = \inf\{\sum_{j=1}^{d_x} \frac{\mathbb{E}_{p_x p_y} \left| \frac{\partial f(x,y)}{\partial x_j} \right|^2 + \varepsilon}{\eta_j} : \eta, \eta_j > 0, \sum_{j=1}^{d_x} \eta_j = 1\},$$

where the optimum is achieved for : $\eta_{j,\varepsilon}^* = \frac{\beta_j}{\sum_{k=1}^{d_x} \beta_k}$, where $\beta_j^2 = \mathbb{E}_{p_x p_y} \left| \frac{\partial f(x,y)}{\partial x_j} \right|^2 + \varepsilon$. We refer to $\eta_{j,\varepsilon}^*$ as the normalized importance score of feature $j$. Note that $\eta_j$ is a distribution over the features and gives a natural ranking between the features. Hence, substituting $\Omega(S)(f)$ with $\Omega_{S,\varepsilon}(f)$ in its equivalent form we obtain the $\varepsilon$ perturbed SIC:

$$\text{SIC}_{(L_1)^2,\varepsilon}(p_{xy}, p_x p_y) = -\inf\{L_\varepsilon(f, \eta) : f \in \mathscr{F}, \eta_j, \eta_j > 0, \sum_{j=1}^{d_x} \eta_j = 1\}$$

where $L_\varepsilon(f, \eta) = -\Delta(f, p_{xy}, p_x p_y) + \frac{\lambda}{2} \sum_{j=1}^{d_x} \frac{\mathbb{E}_{p_x p_y}\left|\frac{\partial f(x,y)}{\partial x_j}\right|^2 + \varepsilon}{\eta_j} + \frac{\rho}{2}\mathbb{E}_{p_x p_y} f^2(x, y)$, and $\Delta(f, p_{xy}, p_x p_y) = \mathbb{E}_{p_{xy}} f(x, y) - \mathbb{E}_{p_x p_y} f(x, y)$. Finally, SIC can be empirically estimated as

$$\widehat{\text{SIC}}_{(L_1)^2,\varepsilon}(p_{xy}, p_x p_y) = -\inf\{\hat{L}_\varepsilon(f, \eta) : f \in \mathscr{F}, \eta_j, \eta_j > 0, \sum_{j=1}^{d_x} \eta_j = 1\}$$

where $\hat{L}_\varepsilon(f, \eta) = -\hat{\Delta}(f, p_{xy}, p_x p_y) + \frac{\lambda}{2} \sum_{j=1}^{d_x} \frac{\frac{1}{N}\sum_{i=1}^{N}\left|\frac{\partial f(x_i,\tilde{y}_i)}{\partial x_j}\right|^2 + \varepsilon}{\eta_j} + \frac{\rho}{2}\frac{1}{N}\sum_{i=1}^{N} f^2(x_i, \tilde{y}_i)$, and main the objective $\hat{\Delta}(f, p_{xy}, p_x p_y) = \frac{1}{N}\sum_{i=1}^{N} f(x_i, y_i) - \frac{1}{N}\sum_{i=1}^{N} f(x_i, \tilde{y}_i)$.

**Remark 2** (Group Sparsity). *We can define similarly nonlinear group sparsity, if we would like our critic to depends on subsets of coordinates. Let $G_k, k = 1, \ldots, K$ be an overlapping or non overlapping group :* $\Omega_{gS}(f) = \sum_{k=1}^{K} \sqrt{\sum_{j \in G_k} \mathbb{E}_{p_x p_y}\left|\frac{\partial f(x,y)}{\partial x_j}\right|^2}$. *The $\eta$-trick applies naturally.*

## 4  Convex Sobolev Independence Criterion in Fixed Feature Spaces

We will now specify the function space $\mathscr{F}$ in SIC and consider in this Section critics of the form:
$$\mathscr{F} = \{f | f(x, y) = \langle u, \Phi_\omega(x, y) \rangle, \|u\|_2 \le \gamma\},$$
where $\Phi_\omega : \mathcal{X} \times \mathcal{Y} \to \mathbb{R}^m$ is a fixed finite dimensional feature map. We define the mean embeddings of the joint distribution $p_{xy}$ and product of marginals $p_x p_y$ as follow: $\mu(p_{xy}) = \mathbb{E}_{p_{xy}}[\Phi_\omega(x, y)]$, $\mu(p_x p_y) = \mathbb{E}_{p_x p_y}[\Phi_\omega(x, y)] \in \mathbb{R}^m$. Define the covariance embedding of $p_x p_y$ as $C(p_x p_y) = \mathbb{E}_{p_x p_y}[\Phi_\omega(x, y) \otimes \Phi_\omega(x, y)] \in \mathbb{R}^{m \times m}$ and finally define the Gramian of derivatives embedding for coordinate $j$ as $D_j(p_x p_y) = \mathbb{E}_{p_x p_y}[\frac{\partial \Phi_\omega(x,y)}{\partial x_j} \otimes \frac{\partial \Phi_\omega(x,y)}{\partial x_j}] \in \mathbb{R}^{m \times m}$. We can write the constraint $\|u\|_2 \le \gamma$ as the penalty term $-\tau \|u\|^2$. Define $L_\varepsilon(u, \eta) = \langle u, \mu(p_x p_y) - \mu(p_{xy}) \rangle + \frac{1}{2}\left\langle u, \left(\lambda \sum_{j=1}^{d_x} \frac{D_j(p_x p_y) + \varepsilon}{\eta_j} + \rho C(p_x p_y) + \tau I_m\right) u \right\rangle$. Observe that :
$$\text{SIC}_{(L^1)^2,\varepsilon}(p_{xy}, p_x p_y) = -\inf\{L_\varepsilon(u, \eta) : u \in \mathbb{R}^m, \eta_j, \eta_j > 0, \sum_{j=1}^{d_x} \eta_j = 1\}.$$

We start by remarking that SIC is a form of gradient regularized maximum mean discrepancy [3]. Previous MMD work comparing joint and product of marginals did not use the concept of nonlinear sparsity. For example the Hilbert-Schmidt Independence Criterion (HSIC) [2] uses $\Phi_\omega(x, y) = \phi(x) \otimes \psi(y)$ with a constraint $\|u\|_2 \le 1$. CCA and related kernel measures of dependence [20, 21] use $L_2^2$ constraints $L_2^2(p_x)$ and $L_2^2(p_y)$ on each function space separately.

**Optimization Properties of Convex SIC** We analyze in this Section the Optimization properties of SIC. Theorem 1 shows that the $\text{SIC}_{(L^1)^2,\varepsilon}$ loss function is jointly strictly convex in $(u, \eta)$ and hence admits a unique solution that solves a fixed point problem.

**Theorem 1** (Existence of a solution, Uniqueness, Convexity and Continuity). *Note that $L(u, \eta) = L_{\varepsilon=0}(u, \eta)$. The following properties hold for the SIC loss:*

*1) $L(u, \eta)$ is differentiable and jointly convex in $(u, \eta)$. $L(u, \eta)$ is not continuous for $\eta$, such that $\eta_j = 0$ for some $j$.*

*2) Smoothing, Perturbed SIC: For $\varepsilon \in (0, 1)$, $L_\varepsilon(u, \eta) = L(u, \eta) + \frac{\lambda}{2}\sum_{j=1}^{d_x} \frac{\varepsilon}{\eta_j}$ is jointly strictly convex and has compact level sets on the probability simplex, and admits a unique minimizer $(u_\varepsilon^*, \eta_\varepsilon^*)$.*

*3) The unique minimizer of $L_\varepsilon(u, \eta)$ is a solution of the following fixed point problem: $u_\varepsilon^* = \left(\lambda \sum_{j=1}^{d_x} \frac{D_j(p_x p_y)}{\eta_j^*} + \rho C(p_x p_y) + \tau I_m\right)^{-1}(\mu(p_{xy}) - \mu(p_x p_y))$, and $\eta_{j,\varepsilon}^* = \frac{\sqrt{\langle u_\varepsilon^*, D_j(p_x p_y)u_\varepsilon^*\rangle + \varepsilon}}{\sum_{k=1}^{d_x} \sqrt{\langle u_\varepsilon^*, D_k(p_x p_y)u_\varepsilon^*\rangle + \varepsilon}}$.*

The following Theorem shows that a solution of the unperturbed SIC problem can be obtained from the smoothed $\text{SIC}_{(L_1)^2,\varepsilon}$ in the limit $\varepsilon \to 0$:

**Theorem 2** (From Perturbed SIC to SIC). *Consider a sequence $\varepsilon_\ell$, $\varepsilon_\ell \to 0$ as $\ell \to \infty$, and consider a sequence of minimizers $(u^*_{\varepsilon_\ell}, \eta^*_\ell)$ of $L_{\varepsilon_\ell}(u, \eta)$, and let $(u^*, \eta^*)$ be the limit of this sequence, then $(u^*, \eta^*)$ is a minimizer of $L(u, \eta)$.*

**Interpretability of SIC.** The following corollary shows that SIC can be written in terms of the importance scores of the features, since at optimum the main objective is proportional to the constraint term. It is to the best of our knowledge the first dependency criterion that decomposes in the sum of contributions of each coordinate, and hence it is an interpretable dependency measure. Moreover, $\eta^*_j$ are normalized importance scores of each feature $j$, and their ranking can be used to assess feature importance.

**Corollary 1** (Interpretability of Convex SIC ). *Let $(u^*, \eta^*)$ be the limit defined in Theorem 2. Define $f^*(x, y) = \langle u^*, \Phi_\omega(x, y) \rangle$, and $\|f^*\|_{\mathscr{F}} = \|u^*\|$. We have that*

$$SIC_{(L^1)^2}(p_{xy}, p_x p_y) = \frac{1}{2}\left(\mathbb{E}_{p_{xy}} f^*(x, y) - \mathbb{E}_{p_x p_y} f^*(x, y)\right)$$

$$= \frac{\lambda}{2}\left(\sum_{j=1}^{d_x}\sqrt{\mathbb{E}_{p_x p_y}|\frac{\partial f^*(x,y)}{\partial x_j}|^2}\right)^2 + \frac{\rho}{2}\mathbb{E}_{p_x p_y} f^{*,2}(x, y) + \frac{\tau}{2}\|f^*\|^2_{\mathscr{F}}.$$

*Moreover, $\sqrt{\mathbb{E}_{p_x p_y}|\frac{\partial f^*(x,y)}{\partial x_j}|^2} = \eta^*_j \Omega_{S,L_1}(f^*)$ and $\sum_{j=1}^{d_x}\eta_j = 1$. The terms $\eta^*_j$ can be seen as quantifying how much dependency as measured by SIC can be explained by a coordinate $j$. Ranking of $\eta^*_j$ can be used to rank influence of coordinates.*

Thanks to the joint convexity and the smoothness of the perturbed SIC, we can solve convex empirical SIC using alternating minimization on $u$ and $\eta$ or block coordinate descent using first order methods such as gradient descent on $u$ and mirror descent [22] on $\eta$ that are known to be globally convergent in this case (see Appendix A for more details).

## 5 Non Convex Neural SIC with Deep ReLU Networks

While Convex SIC enjoys a lot of theoretical properties, a crucial short-coming is the need to choose a feature map $\Phi_\omega$ that essentially goes back to the choice of a kernel in classical kernel methods. As an alternative, we propose to learn the feature map as a deep neural network. The architecture of the network can be problem dependent, but we focus here on a particular architecture: Deep ReLU Networks with biases removed. As we show below, using our sparsity inducing gradient penalties with such networks, results in input sparsity at the level of the witness function $f$ of SIC. This is desirable since it allows for an interpretable model, similar to the effect of Lasso with Linear models, our sparsity inducing gradient penalties result in a nonlinear self-explainable witness function $f$ [23], with explicit sparse dependency on the inputs.

**Deep ReLU Networks with no biases, homogeneity and Input Sparsity via Gradient Penalties.** We start by invoking the Euler Theorem for homogeneous functions:

**Theorem 3** (Euler Theorem for Homogeneous Functions). *A continuously differentiable function $f$ is defined as homogeneous of degree $k$ if $f(\lambda x) = \lambda^k f(x), \forall \lambda \in \mathbb{R}$. The Theorem states that $f$ is homogeneous of degree $k$ if and only if $kf(x) = \langle \nabla_x f(x), x \rangle = \sum_{j=1}^{d_x}\frac{\partial f(x)}{\partial x_j}x_j$.*

Now consider *deep ReLU networks with biases removed* for any number of layers $L$: $\mathscr{F}_{ReLu} = \{f|f(x, y) = \langle u, \Phi_\omega(x) \rangle$, where $\Phi_\omega(x, y) = \sigma(W_L \ldots \sigma(W_2\sigma(W_1[x, y])))$, $u \in \mathbb{R}^m, \Phi_\omega : \mathbb{R}^{d_x+d_y} \to \mathbb{R}^m\}$, where $\sigma(t) = \max(t, 0)$, $W_j$ are linear weights. Any $f \in \mathscr{F}_{ReLU}$ is clearly homogeneous of degree 1. As an immediate consequence of Euler Theorem we then have: $f(x, y) = \langle \nabla_x f(x, y), x \rangle + \langle \nabla_y f(x, y), y \rangle$. The first term is similar to a linear term in a linear model, the second term can be seen as a bias. Using our sparsity-inducing gradient penalties with such networks guarantees that on average on the support of a dominant measure the gradients with respect to $x$ are sparse. Intuitively, the gradients wrt $x$ act like the weight in linear models, and our sparsity inducing gradient penalty act like the $\ell_1$ regularization of Lasso. The main advantage compared to Lasso is that we have a highly nonlinear decision function, that has better capacity of capturing dependencies between $X$ and $Y$.

**Non-convex SIC with Stochastic Block Coordinate Descent (BCD).** We define the empirical non convex $SIC_{(L^1)^2}$ using this function space $\mathscr{F}_{\text{ReLu}}$ as follows:

$$\widehat{\mathrm{SIC}}_{(L^1)^2}(p_{xy}, p_x p_y) = -\inf\{\hat{L}(f_\theta, \eta) : f_\theta \in \mathscr{F}_{\boldsymbol{ReLU}}, \eta_j, \eta_j > 0, \sum_{j=1}^{d_x} \eta_j = 1\},$$

where $\theta = (vec(W_1) \dots vec(W_L), u)$ are the network parameters. Algorithm 3 in Appendix B summarizes our stochastic BCD algorithm for training the Neural SIC. The algorithm consists of SGD updates to $\theta$ and mirror descent updates to $\eta$.

**Boosted SIC.** When training Neural SIC, we can obtain different critics $f_\ell$ and importance scores $\eta_\ell$, by varying random seeds or hyper-parameters (architecture, batch size etc). Inspired by importance scores in random forest, we define **Boosted SIC** as the arithmetic mean or the geometric mean of $\eta_\ell$.

## 6   FDR Control and the Holdout Randomization Test/ Knockoffs.

Controlling the False Discovery Rate (FDR) in Feature Selection is an important problem for reproducible discoveries. In a nutshell, for a feature selection problem given the ground-truth set of features $\mathcal{S}$, and a feature selection method such as SIC that gives a candidate set $\hat{S}$, our goal is to maximize the TPR (True Positive Rate) or the power, and to keep the False Discovery Rate (FDR) under Control. TPR and FDR are defined as follows:

$$\mathrm{TPR} := \mathbb{E}\left[\frac{\#\{i : i \in \hat{S} \cap \mathcal{S}\}}{\#\{i : i \in \mathcal{S}\}}\right] \quad \mathrm{FDR} := \mathbb{E}\left[\frac{\#\{i : i \in \hat{S} \backslash \mathcal{S}\}}{\#\{i : i \in \hat{S}\}}\right]. \tag{1}$$

We explore in this paper two methods that provably control the FDR: 1) The Holdout Randomization Test (HRT) introduced in [8], that we specialize for SIC in Algorithm 4; 2) Knockoffs introduced in [9] that can be used with any basic feature selection method such as Neural SIC, and guarantees provable FDR control.

*HRT-SIC.* We are interested in measuring the conditional dependency between a feature $x_j$ and the response variable $y$ conditionally on the other features noted $x_{-j}$. Hence we have the following null hypothesis: $H_0 : x_j \perp\!\!\!\perp y \,|x_{-j} \iff p_{xy} = p_{x_j|x_{-j}} p_{y|x_{-j}} p_{x_{-j}}$. In order to simulate the null hypothesis, we propose to use generative models for sampling from $x_j|x_{-j}$ (See Appendix D). The principle in HRT [8] that we specify here for SIC in Algorithm 4 (given in Appendix B) is the following: instead of refitting SIC under $H_0$, we evaluate the mean of the witness function of SIC on a holdout set sampled under $H_0$ (using conditional generators for $R$ rounds). The deviation of the mean of the witness function under $H_0$ from its mean on a holdout from the real distribution gives us $p$-values. We use the Benjamini-Hochberg [24] procedure on those $p$-values to achieve a target FDR. We apply HRT-SIC on a shortlist of pre-selected features per their ranking of $\eta_j$.

*Knockoffs-SIC.* Knockoffs [25] work by finding control variables called knockoffs $\tilde{x}$ that mimic the behavior of the real features $x$ and provably control the FDR [9]. We use here Gaussian knockoffs [9] and train SIC on the concatenation of $[x, \tilde{x}]$, i.e we train $SIC([X; \tilde{X}], Y)$ and obtain $\eta$ that has now twice the dimension $d_x$, i.e for each real feature $j$, there is the real importance score $\eta_j$ and the knockoff importance score $\eta_{j+d_x}$. knockoffs-SIC consists in using the statistics $W_j = \eta_j - \eta_{j+d_x}$ and the knockoff filter [9] to select features based on the sign of $W_j$ (See Alg. 5 in Appendix).

## 7   Relation to Previous Work

**Kernel/Neural Measure of Dependencies.** As discussed earlier SIC can be seen as a *sparse* gradient regularized MMD [3, 7] and relates to the Sobolev Discrepancy of [5, 6]. Feature selection with MMD was introduced in [10] and is based on backward elimination of features by recomputing MMD on the ablated vectors. SIC has the advantage of fitting one critic that has interpretable feature scores. Related to the MMD is the Hilbert Schmidt Independence Criterion (HSIC) and other variants of kernel dependency measures introduced in [2, 21]. None of those criteria has a nonparametric sparsity constraint on its witness function that allows for explainability and feature selection. Other Neural measures of dependencies such as MINE [26] estimate the KL divergence using neural networks, or that of [27] that estimates a proxy to the Wasserstein distance using Neural Networks.

**Interpretability, Sparsity, Saliency and Sensitivity Analysis.** Lasso and elastic net [28] are interpretable linear models that exploit sparsity, but are limited to linear relationships. Random forests

[29] have a heuristic for determining feature importance and are successful in practice as they can capture nonlinear relationships similar to SIC. We believe SIC can potentially leverage the deep learning toolkit for going beyond tabular data where random forests excel, to more structured data such as time series or graph data. Finally, SIC relates to saliency based post-hoc interpretation of deep models such as [30, 31, 32]. While those method use the gradient information for a post-hoc analysis, SIC incorporates this information to guide the learning towards the important features. As discussed in Section 2.1 many recent works introduce deep networks with input sparsity control through a learned gate or a penalty on the weights of the network [11, 12, 13]. SIC exploits a stronger notion of sparsity that leverages the relationship between the different covariates.

## 8    Experiments

**Synthetic Data Validation.** We first validate our methods and compare them to baseline models in simulation studies on synthetic datasets where the ground truth is available by construction. For this we generate the data according to a model $y = f(x) + \epsilon$ where the model $f(\cdot)$ and the noise $\epsilon$ define the specific synthetic dataset (see Appendix F.1). In particular, the value of $y$ only depends on a subset of features $x_i$, $i = 1, \ldots, p$ through $f(\cdot)$, and performance is quantified in terms of TPR and FDR in discovering them among the irrelevant features. We experiment with two datasets: **A) Complex multivariate synthetic data (SinExp)**, which is generated from a complex multivariate model proposed in [33] Sec 5.3, where 6 *ground truth* features $x_i$ out of 50 generate the output $y$ through a non-linearity involving the product and composition of the cos, sin and exp functions (see Appendix F.1). We therefore dub this dataset SinExp. To increase the difficulty even further, we introduce a pairwise correlation between all features of 0.5. In Fig. 1 we show results for datasets of 125 and 500 samples repeated 100 times comparing performance of our models with the one of two baselines: Elastic Net (EN) and Random Forest (RF). **B) Liang Dataset.** We show results on the benchmark dataset proposed by [34], specifically the *generalized* Liang dataset matching most of the setup from [8] Sec 5.1. We provide dataset details and results in Appendix F.1 (Results in Figure 2).

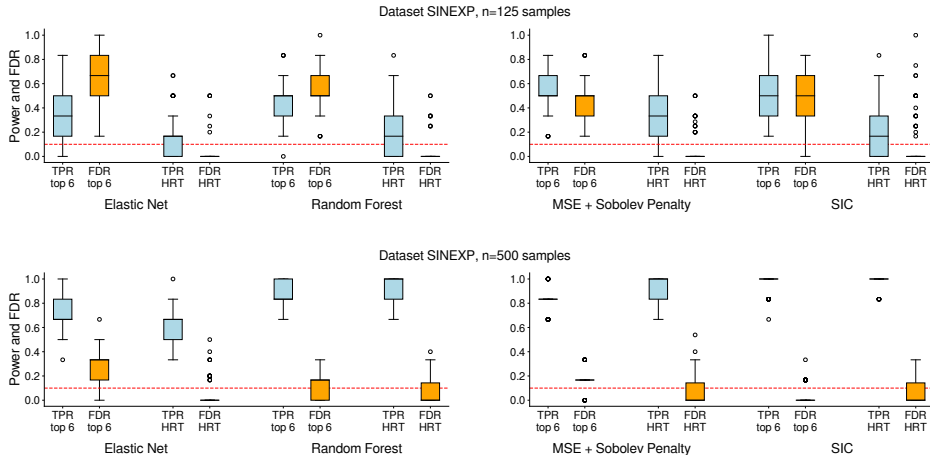

Figure 1: SinExp synthetic dataset. TPR and FDR of Elastic Net (EN) and Random Forest (RF) baseline models (left panels) are compared to our methods: a 2-hidden layer neural network with no biases trained to minimize an objective comprising an MSE cost and a Sobolev Penalty term (MSE + Sobolev Penalty), and the same network trained to optimize SIC criterion (right panels), for datasets of 125 samples (top panels) and 500 samples (bottom panels). For all models TPR and FDR are computed by selecting the top 6 features in order of feature importance (which for EN is defined as the absolute value of the weight of a feature, for RF is the out-of-bag error associated to it (see [35]), and for our method is the value of its $\eta$). Selecting the first 6 features is useful to compare models, but assumes *oracle knowledge* of the fact that there are 6 ground truth features. We therefore also compute FDR and TPR after selecting features using the HRT method of [8] among the top 20 features. HRT estimates the importance of a feature quantifying its effect on the distribution of $y$ on a holdout set by replacing its values with samples from a conditional distribution (see Section 6). We use HRT to control FDR rate at 10% (red horizontal dotted line). Standard box plots are generated over 100 repetitions of each simulation.

**Feature Selection on Drug Response dataset.** We consider as a real-world application the Cancer Cell Line Encyclopedia (CCLE) dataset [36], described in Appendix F.2. We study the result of using the normalized importance scores $\eta_j$ from SIC for (heuristic) feature selection, against features selected by Elastic Net. Table 1 shows the heldout MSE of a predictor trained on selected features, averaged over 100 runs (each run: new randomized 90%/10% data split, NN initialization). The goal here is to quantify the predictiveness of features selected by SIC on its own, without the full randomized testing machinery. The SIC critic and regressor NN were respectively the $big\_critic$ and $regressor\_NN$ described with training details in Appendix F.3, while the random forest is trained with default hyper parameters from scikit-learn [37]. We can see that, with just $\eta_j$, informative features are selected for the downstream regression task, with performance comparable to those selected by ElasticNet, which was trained explicitly for this task. The features selected with high $\eta_j$ values and their overlap with the features selected by ElasticNet are listed in Appendix F.2 Table 3.

| | NN | RF |
|---|---|---|
| All 7251 features | $1.160 \pm 3.990$ | $0.783 \pm 0.167$ |
| Elastic-Net1 [36] top-7 | $0.864 \pm 0.432$ | $0.931 \pm 0.215$ |
| Elastic-Net2 [8] top-10 | $\mathbf{0.663 \pm 0.161}$ | $0.830 \pm 0.190$ |
| SIC top-7 | $0.728 \pm 0.166$ | $0.856 \pm 0.189$ |
| SIC top-10 | $0.706 \pm 0.158$ | $\mathbf{0.817 \pm 0.173}$ |
| SIC top-15 | $0.734 \pm 0.168$ | $0.859 \pm 0.202$ |

Table 1: CCLE results on downstream regression task. Heldout MSE for drug PLX4720 prediction based on selected features. Columns: neural network (NN) and random forest (RF) regressors.

**HIV-1 Drug Resistance with Knockoffs-SIC.** The second real-world dataset that we analyze is the HIV-1 Drug Resistance[38], which consists in detecting mutations associated with resistance to a drug type. For our experiments we use all the three classes of drugs: Protease Inhibitors (PIs), Nucleoside Reverse Transcriptase Inhibitors (NRTIs), and Non-nucleoside Reverse Transcriptase Inhibitors (NNRTIs). We use the pre-processing of each dataset (<drug-class, drug-type>) of the knockoff tutorial [39] made available by the authors. Concretely, we construct a dataset $(X, \tilde{X})$ of the concatenation of the real data and Gaussian knockoffs [9], and fit $SIC([X, \tilde{X}], Y)$. As explained in Section 6, we use in the knockoff filter the statistics $W_j = \eta_j - \eta_{j+d_x}$, i.e. the difference of SIC importance scores between each feature and its corresponding knockoff. For SIC experiments, we use $small\_critic$ architecture (See Appendix F.3 for training details). We use Boosted SIC, by varying the batch sizes in $N \in \{10, 30, 50\}$, and computing the geometric mean of $\eta$ produced by those three setups as the feature importance needed for Knockoffs. Results are summarized in Table 2.

| Drug Class | Drug Type | Knockoff with GLM | | | Boosted SIC Knockoff | | |
|---|---|---|---|---|---|---|---|
| | | TD | FD | FDP | TD | FD | FDP |
| PIs | APV | 19 | 3 | **0.13** | 17 | 5 | 0.22 |
| | ATV | 22 | 8 | 0.26 | 19 | 1 | **0.05** |
| | IDV | 19 | 12 | 0.38 | 15 | 3 | **0.16** |
| | LPV | 16 | 1 | **0.05** | 14 | 2 | 0.12 |
| | NFV | 24 | 7 | 0.22 | 19 | 5 | **0.21** |
| | RTV | 19 | 8 | 0.29 | 12 | 2 | **0.20** |
| | SQV | 17 | 4 | **0.19** | 14 | 8 | 0.36 |
| NRTIs | X3TC | 0 | 0 | 0 | 7 | 0 | **0** |
| | ABC | 10 | 1 | 0.09 | 11 | 1 | **0.08** |
| | AZT | 16 | 4 | **0.2** | 12 | 5 | 0.29 |
| | D4T | 6 | 1 | 0.14 | 8 | 0 | **0** |
| | DDI | 0 | 0 | 0 | 8 | 0 | **0** |
| NNRTIs | DLV | 10 | 13 | 0.56 | 8 | 10 | **0.55** |
| | EFV | 11 | 11 | 0.5 | 11 | 10 | **0.47** |
| | NVP | 7 | 10 | **0.58** | 7 | 11 | 0.611 |

Table 2: Comparison of applying (knockoff filter + GLM) and (Knockoff filter+Boosted SIC). For each <drug-class, drug-type> we compared the True Discoveries (TD), False Discoveries(FD) and False Discovery Proportion (FDP). Knockoff with Boosted SIC keeps FDP under control without compromising power, and succeeds in making true discoveries that GLM with knockoffs doesn't find.

## 9 Conclusion

We introduced in this paper the Sobolev Independence Criterion (SIC), a dependency measure that gives rise to feature importance which can be used for feature selection and interpretable decision making. We laid down the theoretical foundations of SIC and showed how it can be used in conjunction with the Holdout Randomization Test and Knockoffs to control the FDR, enabling reliable discoveries. We demonstrated the merits of SIC for feature selection in extensive synthetic and real-world experiments with controlled FDR.

## Footnotes

*Tom Sercu is now with Facebook AI Research, and Cicero Dos Santos with Amazon AWS AI. The work was done when they were at IBM Research.

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
