[Supplementary Material · SIC_supplement.pdf]

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

# A   Algorithms for Convex SIC

**Algorithms and Empirical Convex SIC from Samples.** Given samples from the joint and the marginals, it is easy to see that the empirical loss $\hat{L}_\varepsilon$ can be written in the same way with empirical feature mean embeddings $\hat{\mu}(p_{xy}) = \frac{1}{N}\sum_{i=1}^N \Phi_\omega(x_i, y_i)$ and $\hat{\mu}(p_x p_y) = \frac{1}{N}\sum_{i=1}^N \Phi_\omega(x_i, \tilde{y}_i)$, covariances $\hat{C}(p_x p_y) = \frac{1}{N}\sum_{i=1}^N \Phi_\omega(x_i, \tilde{y}_i) \otimes \Phi_\omega(x_i, \tilde{y}_i)$ and derivatives grammians $\hat{D}_j(p_x p_y) = \frac{1}{N}\sum_{i=1}^N \frac{\partial \Phi_\omega(x_i, \tilde{y}_i)}{\partial x_j} \otimes \frac{\partial \Phi_\omega(x, y)}{\partial x_j}$. Given the strict convexity of $\hat{L}_\varepsilon$ jointly in $u$ and $\eta$, alternating optimization as given in Algorithm 1 in Appendix is known to be convergent to a global optima (Theorem 4.1 in [40]). Similarly Block Coordinate Descent (BCD) using first order methods as given in Algorithms 3 and 2 (in Appendix): gradient descent on $u$ and mirror descent on $\eta$ (in order to satisfy the simplex constraint [22]) are also known to be globally convergent (Theo 2 in [41].)

---

**Algorithm 1** Alternating Optimization

**Inputs:** $\varepsilon, \lambda, \tau, \rho, \Phi_\omega$
**Initialize** $\hat{\eta}_j = \frac{1}{d_x}, \forall j, \hat{\delta} = \hat{\mu}(p_{xy}) - \hat{\mu}(p_x p_y)$
**for** $i = 1 \ldots$ Maxiter **do**
  $\hat{u} \leftarrow$
  $\left( \lambda \sum_{j=1}^{d_x} \frac{\hat{D}_j(p_x p_y)}{\hat{\eta}_j} + \rho \hat{C}(p_x p_y) + \tau I_m \right)^{-1} \hat{\delta}$

  $\hat{\eta}_j \leftarrow \frac{\sqrt{\langle \hat{u}, \hat{D}_j(p_x p_y)\hat{u}\rangle + \varepsilon}}{\sum_{k=1}^{d_x}\sqrt{\langle \hat{u}, \hat{D}_k(p_x p_y)\hat{u}\rangle + \varepsilon}}$
**end for**
**Output:** $\hat{u}, \hat{\eta}$

---

**Algorithm 2** Block Coordinate Descent

**Inputs:** $\varepsilon, \lambda, \tau, \rho, \alpha, \alpha_\eta$ (learning rates), $\Phi_\omega$
**Initialize** $\hat{\eta}_j = \frac{1}{d_x}, \forall j$, Softmax$(z) = e^z / \sum_{j=1}^{d_x} e^{z_j}$
**for** $i = 1 \ldots$ Maxiter **do**
  Gradient step $u$:
  $\hat{u} \leftarrow \hat{u} - \alpha \frac{\partial \hat{L}_\varepsilon(\hat{u}, \hat{\eta})}{\partial u}$
  Mirror Descent $\eta$ :
  logit $\leftarrow \log(\hat{\eta}) - \alpha_\eta \frac{\partial \hat{L}_\varepsilon(\hat{u}, \hat{\eta})}{\partial \eta}$
  $\hat{\eta} \leftarrow$ Softmax(logit) {stable implementation of softmax}
**end for**
**Output:** $\hat{u}, \hat{\eta}$

---

# B   Algorithms for Neural SIC, HRT-SIC and Model-X Knockoff SIC

---

**Algorithm 3** *(non convex)* Neural SIC$(X, Y)$ (Stochastic BCD )

**Inputs:** $X, Y$ dataset $X \in \mathbb{R}^{N \times d_x}, Y \in \mathbb{R}^{N \times d_y}$, such that $(x_i = X_{i,.}, y_i = Y_{i,.}) \sim p_{xy}$
**Hyperparameters:** $\varepsilon, \lambda, \tau, \rho, \alpha_\theta, \alpha_\eta$ (learning rates)
**Initialize** $\eta_j = \frac{1}{d_x}, \forall j$, Softmax$(z) = e^z / \sum_{j=1}^{d_x} e^{z_j}$
**for** $iter = 1 \ldots$ Maxiter **do**
  Fetch a minibatch of size $N$ $(x_i, y_i) \sim p_{xy}$
  Fetch a minibatch of size $N$ $(x_i, \tilde{y}_i) \sim p_x p_y$
  {$\tilde{y}_i$ obtained by permuting rows of $Y$}
  Stochastic Gradient step on $\theta$:
  $\theta \leftarrow \theta - \alpha_\theta \frac{\partial \hat{L}(f_\theta, \eta)}{\partial \theta}$ {We use ADAM}
  Mirror Descent $\eta$ :
  logit $\leftarrow \log(\eta) - \alpha_\eta \frac{\partial \hat{L}(f_\theta, \eta)}{\partial \eta}$
  $\eta \leftarrow$ Softmax(logit) {stable implementation of softmax}
**end for**
**Output:** $f_\theta, \eta$

---

**Algorithm 4** HRT With SIC $(X, Y)$

**Inputs:** $D_{train} = (X_{tr}, Y_{tr})$ , a Heldout set $D_{\text{Holdout}} = (X, Y)$, features Cutoff $K$
**SIC:** $(f_{\theta^*}, \eta_*) = \text{SIC}(D_{train})$ {Alg. 3}
**Score of witness on Hold out :** $S^* = \text{MEAN}(f_{\theta^*}(X, Y))$
**Conditional Generators** Pre-trained conditional Generator : $G(x_{-j}, j)$ predicts $X_j | X_{-j}$
**Shortlist :** $I = \text{INDEXTOPK}(\eta)$
{p-values for $j \in I$; randomizations tests}
**for** $j \in I$ **do**
  **for** $r = 1 \ldots R$ **do**
    Construct $\tilde{X}$, $\tilde{X}_{.,k} = X_{.,k} \forall k \neq j$ and $\tilde{X}_{.,j} = G(X_{-j}, j)$ {Simulate Null Hyp.}
    $S_{j,r} = \text{MEAN}(f_{\theta^*}(\tilde{X}, Y))$ {Score of witness function on the Null}
  **end for**
  $p_j = \frac{1}{R+1}\left(1 + \sum_{r=1}^R 1_{S_j^r \geq S^*}\right)$
**end for**
discoveries =**BH**(p,targetFDR) {Benjamini-Hochberg Procedure}
**Output:** discoveries

---
**Algorithm 5** Model-X Knockoffs FDR control with SIC
---

**Inputs:** $D_{train} = (X_{tr}, Y_{tr})$, Model-X knockoff features $\tilde{X} \sim$ ModelX($X_{tr}$), target FDR $q$
**Train SIC:** $(f_{\theta^*}, \eta) = \text{SIC}([X_{tr}, \tilde{X}], Y)$, {Alg. 3} where $[X_{tr}, \tilde{X}]$ is the concatenation of $X_{tr}$ and knockoffs $\tilde{X}$
**for** $j = 1, \dots, d_X$ **do**
    Compute importance score of $j$ feature: $W_j = \eta_j - \eta_{j+d_x}$,
    where $\eta_{j+d_x}$ is the $\eta$ of feature knockoff $\tilde{X}_j$
**end for**
Compute threshold $\tau > 0$ by setting
$\tau = \min \left\{ t > 0 : \frac{\#\{j: W_j \leq -t\}}{\#\{j: W_j \geq t\}} \leq q \right\}$
**Output:** discoveries $\{j : W_j > \tau\}$

---

# C Proofs

*Proof of Theorem 1.* 1) Let $\delta = \mu(p_{xy}) - \mu(p_x p_y)$.

We have

$$L(u, \eta) = -\langle u, \delta \rangle + \frac{1}{2} \langle u, (\rho C(p_x p_y) + \tau I_m) u \rangle + \frac{\lambda}{2} \sum_j \frac{\langle u, D_j(p_x p_y) u \rangle}{\eta_j}, u \in \mathbb{R}^m \text{ and } \eta \in \Delta^{d_x}$$

where $\Delta^{d_x}$ is the probability simplex. $L$ is the sum of a linear tem and quadratic terms (convex in $u$) and a function of the form

$$f(u, \eta) = \frac{1}{2} \sum_{j=1}^{d_x} \frac{u^\top A_j u}{\eta_j}$$

where $A_j$ are PSD matrices, and $\eta$ is in the probability simplex (convex). Hence it is enough to show that $f$ is jointly convex. Let $g(w, \eta) = \frac{w^\top A w}{\eta}, \eta > 0$. The Hessian of $g(w, \eta)$, $Hg$ has the following form:

$$Hg(w, \eta) = \begin{bmatrix} \frac{\partial^2 L}{\partial w \otimes \partial w} & \frac{\partial^2 L}{\partial w \partial \eta} \\ \frac{\partial^2 L}{\partial \eta \partial w} & \frac{\partial^2 L}{\partial \eta^2} \end{bmatrix} = \begin{bmatrix} \frac{A}{\eta} & -\frac{Aw}{\eta^2} \\ -\frac{w^\top A}{\eta^2} & \frac{w^\top Aw}{\eta^3} \end{bmatrix}$$

Let us prove that for all $(w, \eta), \eta_j >, \forall j 0$:

$$(w', \eta')^\top Hg(w, \eta)(w', \eta') \geq 0, \forall (w', \eta'), \eta'_j > 0, \forall j$$

We have :

$$
\begin{aligned}
(w', \eta')^\top Hg(w, \eta)(w', \eta') &= \frac{\langle w', Aw' \rangle}{\eta} - 2\eta' \frac{\langle w', Aw \rangle}{\eta^2} + \eta'^2 \frac{w^\top Aw}{\eta^3} \\
&= \frac{1}{\eta} \left( \langle w', Aw' \rangle - \frac{2\eta'}{\eta} \langle w', Aw \rangle + \frac{\eta'^2}{\eta^2} w^\top Aw \right) \\
&= \frac{1}{\eta} \left\| A^{\frac{1}{2}} w' - \frac{\eta'}{\eta} A^{\frac{1}{2}} w \right\|_2^2 \geq 0 \text{ for } \eta > 0
\end{aligned}
$$

Now back to $f$ it is easy to see that :

$$(w', \eta')^\top Hf(w, \eta)(w', \eta') = \sum_{j=1}^{d_x} \frac{1}{\eta_j} \left\| A_j^{\frac{1}{2}} w' - \frac{\eta'_j}{\eta_j} A_j^{\frac{1}{2}} w \right\|_2^2 \geq 0 \text{ for } \eta \in \Delta^{d_x}, \eta_j > 0.$$

Hence the loss $L$ is jointly convex in $(u, \eta)$. Due to discontinuity at $\eta_j = 0$ the loss is not continuous .

2) It is easy to see that the hessian becomes definite:

$$(w', \eta')^\top H L_\varepsilon(w, \eta)(w', \eta') = \sum_{j=1}^{d_x} \frac{1}{\eta_j} \left( \left\| A_j^{\frac{1}{2}} w' - \frac{\eta_j'}{\eta_j} A_j^{\frac{1}{2}} w \right\|_2^2 + \varepsilon(\frac{\eta_j'}{\eta_j})^2 \right) > 0 \text{ for } \eta \in \Delta_j^{d_x}, \eta_j, \eta_j' > 0,$$

and $L_\varepsilon(u, \eta)$ is jointly strictly convex, $u$ is unconstrained and $\eta$ belongs to a convex set (the probability simplex) and hence admits a unique minimizer.

3) The unique minimizer satisfies first order optimality conditions for the following Lagragian:

$$\mathscr{L}(u, \eta, \xi) = L_\varepsilon(u, \eta) + \xi(\sum_j \eta_j - 1)$$

$$\frac{\partial \mathscr{L}(u, \eta, \xi)}{\partial u} = -\delta + \left( \lambda \sum_{j=1}^{d_x} \frac{D_j(p_x p_y)}{\eta_j} + \rho C(p_x p_y) + \tau I_m \right) u = 0$$

and

$$\frac{\partial \mathscr{L}(u, \eta, \xi)}{\partial \eta_j} = -\frac{\lambda}{2} \frac{\langle u, D_j(p_x p_y) u \rangle + \varepsilon}{\eta_j^2} + \xi = 0$$

and

$$\frac{\partial \mathscr{L}(u, \eta, \xi)}{\partial \xi} = \sum_j \eta_j - 1 = 0$$

Hence:

$$u_\varepsilon^* = \left( \lambda \sum_{j=1}^{d_x} \frac{D_j(p_x p_y)}{\eta_j^*} + \rho C(p_x p_y) + \tau I_m \right)^{-1} (\mu(p_{xy}) - \mu(p_x p_y))$$

and :

$$\eta_{j,\varepsilon}^* = \frac{\sqrt{\langle u_\varepsilon^*, D_j(p_x p_y) u_\varepsilon^* \rangle + \varepsilon}}{\sum_{k=1}^{d_x} \sqrt{\langle u_\varepsilon^*, D_k(p_x p_y) u_\varepsilon^* \rangle + \varepsilon}}.$$

$\square$

*Proof of Theorem 2.* The proof follows similar proof in Argryou 2008.

$$S_\varepsilon(u) = L(u_\varepsilon, \eta(u_\varepsilon)) = -\langle u, \delta \rangle + \frac{1}{2} \langle u, (\rho C(p_x p_y) + \tau I_m) u \rangle + \frac{\lambda}{2} \left( \sum_j \sqrt{\langle u, D_j(p_x p_y) u \rangle + \varepsilon} \right)^2$$

Let $\{(u_{\ell_n}, \eta_{\ell_n}(u_{\ell_n})), n \in \mathbb{N}\}$ be a limiting subsequence of minimizers of $L_{\varepsilon_{\ell_n}}(.,.)$ and let $(u^*, \eta^*)$ be its limit as $n \to \infty$. From the definition of $S_\varepsilon(u)$, we see that $\min_u S_\varepsilon(u)$ decreases as $\varepsilon$ decreases to zero, and admits a limit $\bar{S} = \min_u S_0(u)$. Hence $S_{\varepsilon_{\ell_n}} \to \bar{S}$. Note that $S_\varepsilon(u)$ is continuous in both $\varepsilon$ and $u$ and we have finally $S_0(u^*) = \bar{S}$, and $u^*$ is a minimizer of $S_0$. $\square$

*Proof of Corollary 1 .* The optimum $(u_\varepsilon^*, \eta_\varepsilon^*)$ satisfies:

$$-\delta + \left( \lambda \sum_{j=1}^{d_x} \frac{D_j(p_x p_y)}{\eta_j} + \rho C(p_x p_y) + \tau I_m \right) u_\varepsilon^* = 0$$

Let $f^*(x) = \langle u, \Phi_\omega(x,y)\rangle$ and define $||f^*_\varepsilon||_{\mathscr{F}} = ||u^*_\varepsilon||_2$. It follows that $\eta^*_j = \dfrac{\sqrt{\mathbb{E}_{p_x p_y}\left|\frac{\partial f^*_\varepsilon(x,y)}{\partial x_j}\right|^2 + \varepsilon}}{\sum_k \sqrt{\mathbb{E}_{p_x p_y}\left|\frac{\partial f^*_\varepsilon(x,y)}{\partial x_k}\right|^2 + \varepsilon}}$

Note that we have
$$
\begin{aligned}
&\quad \mathbb{E}_{p_{xy}}f^*_\varepsilon(x,y) - \mathbb{E}_{p_x p_y}f^*_\varepsilon(x,y)\\
&= \langle \delta, u^*_\varepsilon\rangle\\
&= \left\langle u^*_\varepsilon, \left(\lambda\sum_{j=1}^{d_x}\frac{D_j(p_x p_y)}{\eta^*_{j,\varepsilon}} + \rho C(p_x p_y) + \tau I_m\right)u^*_\varepsilon\right\rangle\\
&= \lambda\left(\sum_{j=1}^{d_x}\sqrt{\mathbb{E}_{p_x p_y}|\frac{\partial f^*_\varepsilon(x,y)}{\partial x_j}|^2 + \varepsilon}\right)^2 + \rho\mathbb{E}_{p_x p_y}f^{*,2}_\varepsilon(x,y) + \tau||f^*_\varepsilon||^2_{\mathscr{F}}
\end{aligned}
$$

$$
\begin{aligned}
SIC_{(L^1)^2,\varepsilon} &= \mathbb{E}_{p_{xy}}f^*_\varepsilon(x,y) - \mathbb{E}_{p_x p_y}f^*_\varepsilon(x,y) - \frac{1}{2}(\lambda\left(\sum_{j=1}^{d_x}\sqrt{\mathbb{E}_{p_x p_y}|\frac{\partial f^*_\varepsilon(x,y)}{\partial x_j}|^2 + \varepsilon}\right)^2\\
&\quad + \rho\mathbb{E}_{p_x p_y}f^{*,2}_\varepsilon(x,y) + \tau||f^*_\varepsilon||^2_{\mathscr{F}})\\
&= \frac{\lambda}{2}\left(\sum_{j=1}^{d_x}\sqrt{\mathbb{E}_{p_x p_y}|\frac{\partial f^*_\varepsilon(x,y)}{\partial x_j}|^2 + \varepsilon}\right)^2 + \frac{\rho}{2}\mathbb{E}_{p_x p_y}f^{*,2}_\varepsilon(x,y) + \frac{\tau}{2}||f^*_\varepsilon||^2_{\mathscr{F}}\\
&= \frac{1}{2}\left(\mathbb{E}_{p_{xy}}f^*_\varepsilon(x,y) - \mathbb{E}_{p_x p_y}f^*_\varepsilon(x,y)\right)
\end{aligned}
$$

We conclude by taking $\varepsilon \to 0$. $\qquad\square$

## D  FDR Control with HRT and Conditional Generative Models

The Holdout Randomization Test (HRT) is a principled method to produce valid $p$-values for each feature, that enables the control over the false discovery of a predictive model [8]. The $p$-value associated to each feature $x_j$ essentially quantifies the result of a conditional independence test with the null hypothesis stating that $x_j$ is independent of the output $y$, conditioned on all the remaining features $\mathbf{x}_{-j} = (x_1, \ldots, x_{j-1}, x_{j+1}, \ldots, x_p)$. This in practice requires the availability of an estimate of the complete conditional of each feature $x_j$, i.e. of $P(x_j|\mathbf{x}_{-j})$. HRT then samples the values of $x_j$ from this conditional distribution to obtain the $p$-value associated to it. Taking inspiration from neural network models for conditional generation (see e.g. [42]) we train a neural network to act as a generator of a features $x_j$ given the remaining features $\mathbf{x}_{-j}$ as inputs, as a replacement for the conditional distributions $P(x_j|\mathbf{x}_{-j})$. In all of our tasks, one three-layer neural network with 200 ReLU units and Conditional Batch Normalization (BCN) [43] applied to all hidden layers serves as generator for all features $j = 1, \ldots, p$. A sample from $P(x_j|\mathbf{x}_{-j})$ is generated by giving as input to the network an index $j$ indicating the feature to generate, and a sample $\mathbf{x}_{-j} \sim P(\mathbf{x}_{-j})$, represented as a sample from the full joint distribution $\mathbf{x} \sim P(x_1, \ldots, x_p)$, with feature $j$ being masked out. In practice, the index $j$ and $\mathbf{x} \sim P(x_1, \ldots, x_p)$ are given as inputs to the generator, and the neural network model does the masking, and sends the index $j$ to the CBN modules which normalize their inputs using $j$-dependent centering and normalization parameters. The output of the generator is a $n_{bins}$-dimensional softmax over bins tessellating the range of the distribution of $x_j$, such that the bins are uniform quantiles of the inverse CDF of the distribution of $x_j$ estimated over the training set. In all simulations we used a number of bins $n_{bins} = 100$.

Generators are trained randomly sampling an index $j = 1, \ldots, p$ for each sample $\mathbf{x}$ in the training set, and minimizing the cross-entropy loss between the output of the generator neural network $Gen(j, \mathbf{x})$ and $x_j$ using mini-batch SGD. In particular, we used the Adam optimizer [44] with the default pytorch [45] parameters and learning rate $\lambda = 0.003$ which is halved every 20 epochs, and batch size of 128.

# E   Discussion of SIC: Consistency, Computational Complexity and FDR Control

**SIC consistency.**   In order to recover the correct conditional independence we elected to use FDR control techniques to perform those dependent hypotheses testing (btw coordinates). By combining SIC with HRT and knockoffs we can guarantee that the correct dependency is recovered while the FDR is under control. For the consistency of SIC in the classical sense, one needs to analyze the solution of SIC, when the critic is not constrained to belonging to an RKHS. This can be done by studying the solution of the equivalent PDE corresponding to this problem (which is challenging, but we think can also be managed through the $\eta$- trick). Then one would proceed by finding 1) conditions under which this solution exists in the RKHS, 2) generalization bounds from samples to the population solution in the RKHS. We leave this analysis for future work.

**Computational Complexity of Neural SIC.**   The cost of training SIC with SGD and mirror descent has the same scaling in the size of the problem as training the base regressor neural network via back-propagation. The only additional overhead is the gradient penalty, where the cost is that of double back-propagation. In our experiments, this added computational cost is not an issue when training is performed on GPU.

**SIC-HRT versus SIC-Knockoffs.**   For a comparison between HRT and knockoffs, we refer the reader to [8], which shows similar performance for either method in terms of controlling FDR. Each method has its advantages. In HRT most of the computation is in 1) training the generative models, and 2) performing the randomization test, i.e. forwarding the data through the critic and computing $p$-values for each coordinate for $R$ runs. On the other hand, if knockoff features can be modelled as a multivariate Gaussian, controlling FDR with knockoffs can be done very cheaply, since it does not require randomization tests. If instead knockoff features have to be generated through nonlinear models, knockoffs can be computationally expensive as well (see for example [46]).

# F   Experimental details

## F.1   Synthetic Datasets

### F.1.1   Complex Multivariate Synthetic Dataset (SinExp)

The SinExp dataset is generated from a complex multivariate model proposed in [33] Sec 5.3, where 6 features $x_i$ out of 50 generate the output $y$ through a non-linearity involving the product and composition of the $\cos$, $\sin$ and $\exp$ functions, as follows:

$$y = \sin(x_1(x_1 + x_2)) \cos(x_3 + x_4 x_5) \sin(e^{x_5} + e^{x_6} - x_2).$$

We increase the difficulty even further by introducing a pairwise correlation between all features of 0.5. We perform experiments using datasets of 125 and 500 samples. For each sample size, 100 independent datasets are generated.

### F.1.2   Liang Dataset

*Liang Dataset* is a variant of the synthetic dataset proposed by [34]. The dataset prescribes a regression model with 500-dimensional correlated input features $x$, where the 1-D regression target $y$ depends on the first 40 features only (the last 460 correlated features are ignored). In the original dataset proposed by [34], $y$ depends on 4 features only, this more complex version of the dataset that uses 40 features was proposed by [8]. The target $y$ is computed as follows:

$$y = \sum_{j=0}^{9} [w_{4j} x_{4j} + w_{4j+1} x_{4j+1} + \tanh(w_{4j+2} x_{4j+2} + w_{4j+3} x_{4j+3})] + \sigma \epsilon \,, \tag{2}$$

with $\sigma = 0.5$ and $\epsilon \sim \mathcal{N}(0, 1)$. As in [8], the 500 features are generated to have 0.5 correlation coefficient with each other,

$$x_j = (\rho + z_j)/2 \,, \quad j = 1, \ldots, 500 \,, \tag{3}$$

where $\rho$ and $z_j$ are independently generated from $\mathcal{N}(0, 1)$.

Our experimental results are the average over 100 generated datasets, each consisting of 500 train and 500 heldout samples.

Figure 2: Liang synthetic dataset. TPR and FDR of Elastic Net baseline models (left panels) are compared against our methods, analogously to Fig. 1. Differently from Fig. 1, however, TPR and FDR are computed by selecting the top 40 features in order of importance (since this dataset was generated with 40 ground truth features). Moreover, HRT is used to select features among the top 100 most important features.

## F.2 CCLE Dataset

The Cancer Cell Line Encyclopedia (CCLE) dataset [36] provides data about of anti-cancer drug response in cancer cell lines. The dataset contains the phenotypic response measured as the area under the dose-response curve (AUC) for a variety of drugs that were tested against hundreds of cell lines. [36] analyzed each cell to obtain gene mutation and expression features. The total number of data points (cells) is 479. We followed the preprocessing steps by [8] and first screened the genomic features to filter out features with less than 0.1 magnitude Pearson correlation to the AUC. This resulted in a final set of about 7K features. The main goal in this task is to discover the genomic features associated with drug response. Following [8], we perform experiments for the drug PLX4720. Table 3 presents the top-10 genomic features selected by SIC according to $\eta_j$ values. In Sec. 8, we also present quantitative results that show the effectiveness of these features when used to train regression models.

|   | Genomic Feature | $\eta_j$ |
|---|---|---|
| 0 | BRAF.V600E_MUT * | 0.011837 |
| 1 | ACKR3 | 0.011712 |
| 2 | RP11-349I1.2 | 0.010534 |
| 3 | BRAF_MUT † | 0.010449 |
| 4 | UBE2V1P5 | 0.010420 |
| 5 | EPB41L3 | 0.010163 |
| 6 | C11orf85 † | 0.009622 |
| 7 | RP11-395F4.1 | 0.009449 |
| 8 | SERPINA9 | 0.009387 |
| 9 | RN7SKP281 | 0.009369 |

Table 3: Top-10 Genomic Features selected by SIC according to $\eta_j$ values. These are the most important features for high mutual information with PLX4720 response variable, on the CCLE dataset. * indicates feature also discovered by Elastic Net and HRT [8]. † indicates feature also discovered by Elastic Net in original CCLE paper [36].

## F.3 SIC Neural Network descriptions and training details

The first critic network used in the experiments (with SinExp and HIV-1 datasets) is a standard three-layer ReLU dropout network with no biases, i.e. small_critic. When using this network, the

inputs $X$ and $Y$ are first concatenated then given as input to the network. The two first layers have size 100, while the last layer has size 1. We train the network using Adam optimizer with $\beta_1 = 0.5$, $\beta_2 = 0.999$, weight_decay=1e-4 learning rate $\alpha_\eta$ = 1e-3 and $\alpha_\eta = 0.1$, and perform 4000 training iterations/updates, computed with batches of size 100. All NNs used in our experiments were implemented using PyTorch [45].

```
small_critic(
  (branchxy): Sequential(
    (0): Linear(in_features=51, out_features=100, bias=False)
    (1): ReLU()
    (2): Dropout(p=0.3)
    (3): Linear(in_features=100, out_features=100, bias=False)
    (4): ReLU()
    (5): Dropout(p=0.3)
    (6): Linear(in_features=100, out_features=1, bias=False)
  )
)
```

The critic network used in the experiments with Liang and CCLE datasets contains two different branches that separately process the inputs $X$ (*branchx*) and $Y$ (*branchy*), then the output of these two branches are concatenated and processed by a final branch that contains three-layer LeakyReLU network (*branchxy*). We name this network *big_critic* (see figure bellow for details about layer sizes). This network is trained with the same Adam settings as above for 4000 updates (Liang) and 8000 updates (CCLE).

```
big_critic(
  (branchx): Sequential(
    (0): Linear(in_features=500, out_features=100, bias=True)
    (1): LeakyReLU(negative_slope=0.01)
    (2): Linear(in_features=100, out_features=100, bias=True)
    (3): LeakyReLU(negative_slope=0.01)
  )
  (branchy): Sequential(
    (0): Linear(in_features=1, out_features=100, bias=True)
    (1): LeakyReLU(negative_slope=0.01)
    (2): Linear(in_features=100, out_features=100, bias=True)
    (3): LeakyReLU(negative_slope=0.01)
  )
  (branchxy): Sequential(
    (0): Linear(in_features=200, out_features=100, bias=True)
    (1): LeakyReLU(negative_slope=0.01)
    (2): Linear(in_features=100, out_features=100, bias=True)
    (3): LeakyReLU(negative_slope=0.01)
    (4): Linear(in_features=100, out_features=1, bias=True)
  )
)
```

The regressor NN used for the downstream regression task in Section 8 is a standard three-layer ReLU dropout network. This regressor NN was trained with the same Adam settings as above for 1000 updates with a batchSize of 16. We did not perform any hyperparameter tuning or model selection on heldout MSE performance.

```
regressor_NN(
  (net): Sequential(
    (0): Linear(in_features=7251, out_features=100, bias=True)
    (1): ReLU()
    (2): Dropout(p=0.3)
    (3): Linear(in_features=100, out_features=100, bias=True)
    (4): ReLU()
    (5): Dropout(p=0.3)
```

```
    (6): Linear(in_features=100, out_features=1, bias=True)
  )
)
```