[Reviews · NeurIPS 2019]

Reviewer 1



Significance: The proposed criterion (SIC) relies on maximizing a discrepancy between a joint distribution and the product of its marginals over a set of test functions. Unlike existing criterions, SIC incorporates a gradient constraint on such test functions which enforces sparsity in the coordinates. This provides a way of selecting the most dependent dimensions for which the gradient of the optimal test function will not vanish. Such a criterion provides with additional information compared to the mutual Information or to the HSIC. In that sense, it has potentially many applications. Originality: The idea of using a sparsity-inducing penalty on the gradient of the test functions leads to an elegant formulation that provides interpretable scores of dependence on each variable. Clarity: Very clear presentation of the method and good organization of the paper. I have one minor suggestion: I found, Figure 1 hard to parse at first because it makes it hard to compare the different methods: TPR and FDR are expected to behave differently (TPR: highest is better and opposite for FDR) but they are all in the same figures. Would it be better to have one figure for TPR and another one for FDR? Quality: The paper is technically sound and precise. The experiments seem to support the main claims of the paper which is to learn the dimensions that maximize the dependence between variables. The authors compared the proposed method with other methods: Elastic net, Random forest and simple MSE with an additional gradient penalty. It seems that overall, SIC performs as well as random forests on simple examples but benefits from additional interpretability. It also seems to lead to better False discovery proportions on more complicated datasets compared to GLMs. However, I still have a few questions about the paper: It is not totally clear to me how the gradient penalty could help more than simply using the formulation (P) which learns a sparse selector gate to detect the most relevant features. I think it would be very beneficial to the paper to include an experimental comparison with this simple formulation (for instance in figure 1). Also in the convex case, the authors show the existence of the solution but what remains open is the consistency of the estimator obtained in that case: does it recover the true independence structure? Finally, in section 5, a neural network without bias was used which leads to a class of homogeneous functions. Such class can be very restrictive in that it doesn't have universall approximation capabilities. How critical is this choice of function for the method? Overall I think that the paper proposes an interesting and novel method that has many applications. ================================ After reading the other reviews and the authors' response, I still think this is a good submission and should be accepted. It would be great to incorporate the explanations provided in the response to the final version of paper.

Reviewer 2



Updated review: I thank the authors for their effort and time in addressing the authors concerns. I am happy with the provided clarifications and will keep my score as is. --------------------------- In this paper, the authors tackle the important problem of feature selection. Moreover, they focus on deriving an interpretable feature selection method that can easily be combined to control the rate of false discovery. Both having an interpretable feature selection method as well as efficiently controlling the false discovery rate are important problems in themselves. This paper proposes the Sobolev Independence Criterion, a criterion combining aspects of integral probability measures and gradient-sparsity penalties. The authors do a very good job in motivating the gradient sparsity regularization for SIC. While the first proposal of SIC is non-smooth and biased, auxiliary variables are introduced to mitigate the problem. Furthermore, the auxiliary variables have the convenient interpretation of being normalized importance scores for the features. Under a special class of critics (as considered in section 4), the authors show that SIC is convex and they prove the existence of a unique solution and the convergence of the perturbed SIC to the unperturbed SIC in the limit. To the best of my knowledge, the proofs provided in the appendix are correct. Next, the authors combine SIC with deep relu networks and define the Boosted SIC using different random seeds. As an application of SIC false discovery rate control is proposed, and SIC is combined with holdout randomization tests and knockoffs. While the related work section is rather short, it is succinct and mentions the important previous work in the area. To improve this part of the paper a bit, I would propose to include a bit more detailed comparison of the proposed method with previous work (e.g. what shortcomings of previous methods are mitigated with SIC?). At the end, an experimental evaluation of SIC is provided on both synthetic and real-world datasets. While not outperforming the competing methods on all instances of all datasets, SIC shows competitive performance. While there is a large diversity in the datasets considered, there is only comparison to 1 competing method in the two real-world datasets. I would advise the authors to provide comparison with further competing methods to strengthen the case for their approach. Overall, this paper is very nicely and coherently written with a nice balance of motivation and details in the main text with the proofs present in the appendix. Furthermore, this paper makes as interesting and novel contribution to the field of feature selection with convincing experimental results. Thus, I recommend the acceptance of this paper.

Reviewer 3



Summary: The paper proposed the Sobolev Independence Criterion (SIC), an interpretable dependency measure between multivariate random variables X (input) and Y (response). It is an integral probability metric between the joint distribution and the product of the marginals, regularised by gradient of the witness function wrt each dimensions of X. For a general function space, the optimisation problem has l1- like penalty on the gradient terms to ensure sparsity and l2-like penalty on the square of the witness function to induce smoothness. This form of the optimisation is not easy to work with: 1) expectation appears after the square root in the gradient penalties (i.e. a non-smooth term) and 2) the expectation inside the nonlinearity introduces a gradient estimation bias (biased expectation estimation). The authors proposed to alleviated these problem by introducing an auxiliary variable \eta through the variational form of the square root. This \eta_j parameter is the normalised importance score for each feature j of X and hence can be used for feature selection through their ranking. Two classes of functions have been studied in the paper: fixed feature space and deep ReLu networks with biases removed. When the SIC witness function f is restricted to an RKHS which lead to an optimisation problem that is jointly convex in f and \eta, they provide theoretical guarantees on the existence and uniqueness of the solution [Theorem 1], convergence results of the perturbed SIC (i.e. SIC_\epsilon where \epsilon is an regularisation term added inside the square root of the nonlinear sparsity penalty to induce smoothness) to SIC [Theorem 2] and the decomposition of SIC into the sum of contributions from each coordinates [Corollary 1]. Instead of choose a feature map, they propose to use deep ReLu networks to learn them, however the optimisation problem becomes non convex. They utilises existing optimisation algorithms and FDR control methods to establish SIC based feature selection algorithms. The paper concludes with experiments on two synthetic dataset and two real life datasets. The paper is clearly written however there are a few typos. The structure is logical and explanation is adequate. It utilises the sparsity inducing gradient penalties, that were popular in other areas of machine learning (e.g. double back-propagation, WGAN_GP, optimal transport theory, etc.), as a regularisation term in the traditional integral probability metric. Consequently, one has direct access to the importance score of each feature and this naturally leads to a feature selection scheme. Such formulation is general and one is allowed to consider different function spaces. The work provides an interesting formulation of a dependency measure and of the problem of feature selection, which is worth exploring further. However, overall I find the experiments section requires much more discussion, or at times, more experiments to illustrate the performance of the proposed procedure. I have a few questions/concerns that require some clarifications from the authors: Main concerns: 1) From my understanding, the paper proposed two FDR controlled SIC: HRT-SIC and knockoff SIC. Some more experiments or explanations might be needed since it is not clear to me if the two methods perform similarly? Or are there situations where one clearly is better than the other? Or there are situations where one can not apply one of them? 2) The authors proposed the a kernel SIC and then alleviate the problem of kernel selection by introducing the neural SIC, where the feature maps are learnt. My understanding is that it is highly problem/data dependent when neural networks outperforms kernel methods. Have you tried to use a Gaussian kernel with a fix length-scale (e.g. median heuristic) in the experimental comparison? Is it always better to use the neural SIC? I feel some discussion/experiments are missing here. 3) There is no discussion of the trade-off between the computational time and performance. Can the authors provide some comments on this? I find it hard to judge the gain in the performance without knowing the computational cost of the proposed method. 4) I am a little confused by the beginning of section 4: this seems to be a slightly different formulation from the usual kernel set up. For HSIC (or MMD), the kernel is required to be bounded (i.e. E_z(\sqrt{k(z,z)})< \infty) for the existence of the kernel mean embedding and for the mean embedding to be injective, we further require the kernel to be characteristic. The only constraint on this space of function seems to be through the parameter u, does it mean we can have any feature map \Phi_\omega in the setting presented (e.g. tensor product of feature maps that correspond to non characteristic kernels?) Or does these requirements translates into condition on the parameter u? Or is it in fact not necessary to have any conditions on the feature map? Can the authors please explain these? 5) line 116 on page 4, Theorem 1, (2) why is it important to show that L_\epsilon has compact level sets on the probability simplex? It is also not clear to me how it was show in the proof? Minor concerns: 1) line 49 on page 2, the authors mentioned that it becomes a generalised definition of Mutual information, has this “generalised” concept been studied somewhere already? 2) line 55 equation, what is this zero norm of w? 3) line 90 on page 4, why can we remove the 1/N term from the last term? 4) The HIV-1Drug Resistance with Knockoffs-SIC experiment, it is not clear to me if it is a single run of the experiments or an average over multiple runs? What does GLM stands for? 5) line 386 on page 13, it is not clear to me how we can take partial derivative of L, we can obtain the matrix on the right? Should it not be g? Also the line before line 387 has some typos. ========================= Authors comments read. I am happy with the comments provided by the authors, it would be great if the relevant explanations provided could be elaborated in the revision. I have adjusted my score accordingly.

[Author Response · NeurIPS 2019]

1 We thank the reviewers for their valuable and positive feedback. We will address their main concerns:

2 **Reviewer #1.** We thank the reviewer for the positive and encouraging feedback and address their suggestions and questions: *1) On Clarity "TPR and FDR same Figure":* For space restriction, and in line with prior papers eg HRT [8], we elected to have them in the same plot, but we will make sure to remind the reader of the expected trends in the legend of the figure. *2) Comparison to the Gate Formulation in (P):* In our early experiments using a NN critic, we found this formulation to be under-performing. Theoretically, Formulation $(P)$ does not lead to a convex formulation in the RKHS. (P) was indeed also studied in the cited reference [10] (Appendix A), using the Hilbert-Schmidt Independence Criterion and a concave approximation of $\ell_0$, and was found to be under-performing as well. *3) "Analysis of the Consistency":* This is an interesting question. In order to recover the correct conditional independence we elected to use FDR control techniques to perform those dependent hypotheses testing (btw coordinates). By combining SIC with HRT and knockoffs we can guarantee that the correct dependency is recovered while the FDR is under control. For the consistency of SIC in the classical sense, one needs to analyze the solution of SIC, when the critic is not constrained to belong to an RKHS. This can be done by studying the solution of the equivalent PDE corresponding to this problem (which is challenging, but we think it can be also managed through the $\eta-$ trick). Then one would proceed by finding 1) conditions under which this solution exists in the RKHS 2) Generalization bounds from samples to the population solution in the RKHS. We can mention these points in the final version, but we feel that a thorough analysis is beyond the scope of the paper and will be left for future work. *4) "ReLu, No Biases":* The choice of Relu with no biases is dictated by requiring interpretability in terms of "input sparsity" of the neural network. In practice, there is no restriction on the network and one can use Relu with biases if input sparsity is not a goal. In any case, removing biases in the ReLU network does not seem to affect performance strongly, especially when the input data is centered.

21 **Reviewer #2.** We thank the reviewer for the positive and encouraging feedback and address their main concerns here. **Additional Comparison on the real word datasets:** For the CCLE experiments, we compared to ElasticNet which is the established feature selection method on this dataset [8, 36]. For the HIV knockoff experiments, we compared to the GLM feature selection of Candes et al [9]. We plan to include the Random Forest feature selections in the final version as an additional baseline, as suggested by the reviewer.

26 **Reviewer #3.** We thank the reviewer for the review and address their questions: *1) HRT versus Knockoffs:* Thank you for pointing this out and allowing us to clarify that this paper is not a comparative study between the two methods, rather it is a strength of SIC that it can be used with both. For a comparison between HRT and knockoffs, we refer the reviewer to [8], which shows similar performance for either method in terms of controlling FDR. We will highlight in the final paper that each method has its advantages. In HRT most of the computation is in 1) training the generative models, and 2) performing the randomization test, i.e. forwarding the data through the critic and computing $p$-values for each coordinate for $R$ runs. On the other hand, if knockoff features can be modelled as multivariate Gaussians, controlling FDR with knockoffs can be done very cheaply, since it does not require randomization tests. If instead knockoff features have to be generated through non-linear models, knockoffs can be computationally expensive as well (for example Deep knockoffs, Romano et al. ICLR 2019). *2) Kernel versus NN SIC:* In early experiments, we tried using random networks defined by random Fourier features (approximating a Gaussian kernel), and we found that a fully trained network outperforms the fixed network. *3) Computational Complexity versus Performance:* We thank the reviewer for the question, and plan to clarify this point as follows. The cost of training SIC with SGD and mirror descent is of the same order of magnitude as training the base regressor neural network via back-propagation. The only additional overhead is the gradient penalty, where the cost is for a double back-propagation. In our experiments, this added computational cost is not an issue when training is performed on GPU. *4) Conditions on $\Phi$:* In our framework, $\Phi$ is defined on $\mathcal{X} \times \mathcal{Y}$, but is not restricted to be an outer product between feature maps. It is possible to ensure that $SIC(p_{xy}, p_x p_y) = 0$ iff $p_{xy} = p_x p_y$ by imposing universality on the feature map $\Phi$ to get injectivity of the mean embeddings. However, studying the conditions on the kernels under which $\eta_j = 0$ leads to $x_j \perp y|x_{-j}$ is an interesting open question, but beyond the scope of this paper. Instead, we tackled the problem (whether $\eta_j$ are statistically significantly above zero) indirectly using HRT and Knockoffs that have theoretical guarantees to control the FDR with no restriction on the feature selection method used. *5) Compact level sets of $L_\epsilon$:* This condition is met when perturbing the problem with $\epsilon$. This in not needed for the proof of Theorem 1, but is needed to guarantee that alternating optimization or first order methods on $u$ and $\eta$ are convergent (See the monograph [19] page 59). We address the minor comments: *1) Generalized MI.* There are many generalizations of mutual information (MI) such as the Renyi MI that uses Renyi divergence and many other extensions have been developed. For an introduction on this topic we refer the reviewer to $\alpha$ **mutual information** by Sergio Verdu. *2-3) "zero norm of $w$, $\frac{1}{N}$"* $\ell_0$ norm and $\frac{1}{N}$ should be there. Thanks, typos will be corrected. *4) Multiple runs? GLM?:* This is for a single run, following the same experimental protocol as Candes et al. [9]. GLM stands for Generalized Linear Model. Boosted SIC is an ensemble of $\eta$ for the same data, different random seeds initialization of the NN. **"$L$ versus $g$"** In the cost $L$ the only non-obvious terms for proving convexity in $\eta$ and $u$ are of the form $g(u, \eta_j) = \frac{u^\top A_j u}{\eta_j}$ ($A_j$ PD) hence the proof is here only for those terms.

[Meta-Review · NeurIPS 2019]

This paper proposes Sobolev Independence Criterion for feature selection by sparsity inducing penalty with the integral probability metric. The proposed method is novel and of significance. The paper is very well written. We recommend acceptance of this submission.